# Structure and Properties of Polyoxymethylene/Silver/Maleic Anhydride-Grafted Polyolefin Elastomer Ternary Nanocomposites

**DOI:** 10.3390/polym13121954

**Published:** 2021-06-11

**Authors:** Yang Liu, Xun Zhang, Quanxin Gao, Hongliang Huang, Yongli Liu, Minghua Min, Lumin Wang

**Affiliations:** 1Key Laboratory of Oceanic and Polar Fisheries, Ministry of Agriculture and Rural Affairs, East China Sea Fisheries Research Institute, Chinese Academy of Fishery Sciences, Shanghai 200090, China; liuy133787373118@126.com (Y.L.); zhangxun@ecsf.ac.cn (X.Z.); gaoqx2008@163.com (Q.G.); ecshhl@163.com (H.H.); 1981-lyl@163.com (Y.L.); 2Joint Laboratory for Open Sea Fishery Engineering, Qingdao National Laboratory for Marine Science and Technology, Qingdao 266237, China; 3Hunan Engineering Research Center for Rope & Net, Hunan Xinhai Co., Ltd., Yiyang 413100, China; 4College of Life Science, Huzhou University, Huzhou 313000, China

**Keywords:** POM, POM/Ag nanocomposites, POM/Ag/MAH-*g*-POE ternary nanocomposites, mechanical properties, thermal stability, dynamic mechanical analysis

## Abstract

In the present study, silver (Ag) nanoparticles and maleic anhydride-grafted polyolefin elastomer (MAH-*g*-POE) were used as enhancement additives to improve the performance of the polyoxymethylene (POM) homopolymer. Specifically, the POM/Ag/MAH-*g*-POE ternary nanocomposites with varying Ag nanoparticles and MAH-*g*-POE contents were prepared by a melt mixing method. The effects of the additives on the microstructure, thermal stability, crystallization behavior, mechanical properties, and dynamic mechanical thermal properties of the ternary nanocomposites were studied. It was found that the MAH-*g*-POE played a role in the bridging of the Ag nanoparticles and POM matrix and improved the interfacial adhesion between the Ag nanoparticles and POM matrix, owing to the good compatibility between Ag/MAH-*g*-POE and the POM matrix. Moreover, it was found that the combined addition of Ag nanoparticles and MAH-*g*-POE significantly enhanced the thermal stability, crystallization properties, and mechanical properties of the POM/Ag/MAH-*g*-POE ternary nanocomposites. When the Ag/MAH-*g*-POE content was 1 wt.%, the tensile strength reached the maximum value of 54.78 MPa. In addition, when the Ag/MAH-*g*-POE content increased to 15wt.%, the elongation at break reached the maximum value of 64.02%. However, when the Ag/MAH-*g*-POE content further increased to 20 wt.%, the elongation at break decreased again, which could be attributed to the aggregation of excessive Ag nanoparticles forming local defects in the POM/Ag/MAH-*g*-POE ternary nanocomposites. Furthermore, when the Ag/MAH-*g*-POE content was 20 wt.%, the maximum decomposition temperature of POM/Ag/MAH-*g*-POE ternary nanocomposites was 398.22 °C, which was 71.39 °C higher than that of pure POM. However, compared with POM, the storage modulus of POM/Ag/MAH-*g*-POE ternary nanocomposites decreased with the Ag/MAH-*g*-POE content, because the MAH-*g*-POE elastomer could reduce the rigidity of POM.

## 1. Introduction

Polyoxymethylene (POM) is one of the fastest growing engineering semicrystalline thermoplastics in the world owing to its high tensile strength, high rigidity, low friction coefficient, and impact, thermal, chemical, and solvent resistance [1,2,3]. It is widely used in the areas of automotive, electrical, electronics, and packaging applications [4,5,6]. However, POM has been found to be too weak under impact to be used commercially, attributed to its gap-sensitivity and low impact toughness [7].

One way to toughen polymers is to incorporate various additives, and there has been extensive research regarding the additives modification of POM [8,9,10]. For example, nano silica and polylactic acid-grafted polyethylene glycol (PELA) were used as enhancement additives to improve the performance of the POM homopolymer by Nguyen [11]. The results showed that the tensile strength of POM/PELA/NS nanocomposites increased 15.90% relative to neat POM when 2 wt.% PELA was used. In addition, Durmus investigated the effects of methyl-polyhedral oligomeric silsesquioxanes (methyl-POSS) on the isothermal crystallization behavior of POM [12]. It was found that the reciprocal of crystallization half-time (*τ*_0.5_) of POM (0.133 min^−1^) approximately increased six-fold, when the addition content of methyl-POSS was 2 wt.%. This result demonstrated that methyl-POSS accelerated the crystallization rate of POM. At the same time, Zhao studied the effect of multiwalled carbon nanotubes (MWCNTs) on the crystallization behavior of POM [13]. They pointed out that MWCNTs reduced the induction time of crystallization and improved the crystal growth rate and crystallization temperature of POM. Furthermore, Kuciel determined the influence of fiber geometry on the mechanical behavior of fiber-reinforced POM-based composites of various diameters [14]. The results showed that the addition of up to 10 wt.% fiberglass increased the tensile properties and impact strength more than twice, and the ability to absorb energy also increased in relation to neat POM.

On the one hand, the use of nanocomposites based on silver (Ag) nanoparticles has been one of the dominant trends in the modification of engineering materials over the last few years [15]. Regarding Ag itself, there are several important reasons for blending this material with other polymers. The first one is their unique volume effect and quantum size [16], which act as a multifunctional platform of nanocomposites. Secondly, Ag nanoparticles have gained extensive attention due to their good application prospects in biomedicine, electronics, optics, catalysis, sensors, and life sciences, resulting in their high conductivity, excellent catalytic performance, and broad spectrum of antimicrobial activities [17,18,19,20]. One of the interesting studies was that of the effect of polyethylene (PE)/Ag nanocomposites on the Ag ion release and antimicrobial properties by Zapata [21]. It was found that PE/Ag nanocomposites of higher Ag nanoparticle concentrations (5 wt.%) showed the highest Ag ion release and reached 99.99% efficacy against the bacteria after 24 h. In addition, Rozilah produced antibacterial sugar palm starch biopolymer composite films by the solution casting method and determined the mechanical and physicochemical properties [22]. They pointed out that when the addition of Ag NPs was 3 wt.%, the highest elongation of the biopolymer composites film to rupture was 415%, and the tensile strength increased to 408 kPa. On the other hand, polyolefin elastomer (POE) is a thermoplastic elastomer with uniform short-chain branches [23]. POE has excellent toughness and good processability owing to its unique molecular structure and narrow molecular weight distribution [24]. Therefore, the impact strength of polymers could be improved significantly by means of melt blending with POE [25]. However, this resulted in a decline in tensile and flexural strength of polymers [26].

In order to develop an improved, balanced performance of POM-based nanocomposites, in this work, the Ag nanoparticles were successfully synthesized, and the POM/Ag nanocomposites and POM/Ag/MAH-*g*-POE nanocomposites were then prepared by a simple melt compounding route. The effects of the additives on the microstructure, thermal stability, crystallization behavior, mechanical properties, and dynamic mechanical thermal properties of the ternary nanocomposites were studied.

## 2. Experimental

### 2.1. Materials

Polyoxymethylene (POM) was obtained from Yunnan Yuntianhua Co., Ltd., Shanghai, China. Maleic anhydride-grafted polyolefin elastomer (MAH-*g*-POE) was obtained by Nanjing Plastic Thai Polymer Technology Co., Ltd, Nanjing, China. Silver nitrate (99.8%), oleic acid, ethanol, n-propylamine, ascorbic acid (99.7%), and n-heptane were purchased from Sinopharm Chemical Reagent Co., Ltd., Shanghai, China. All the reagents employed in this work were of analytical grade and were used as received. Aqueous solutions were prepared using deionized water.

### 2.2. Synthesis of Ag Nanoparticles

The synthesis of Ag nanoparticles followed previously published recipes [27]. Briefly, the silver nitrate was reduced by ascorbic acid under the stability of oleic acid and n-propylamine. In addition, the detailed preparation process and characterization of the Ag nanoparticles can be found in the previously published papers [27,28].

### 2.3. Preparation of POM/Ag and POM/Ag/MAH-*g*-POENanocomposites

The preparation of POM/Ag nanocomposites followed previously published recipes [8,9].In addition, for the POM/Ag/MAH-*g*-POE ternary nanocomposites, Ag nanoparticles and MAH-*g*-POE particles were first mixed to obtain MAH-*g*-POE/Ag nanocomposites, and the MAH-*g*-POE/Ag nanocomposites and POM were then mixed to prepare POM/Ag/MAH-*g*-POE ternary nanocomposites, as shown in Figure 1.Before the melt processing, POM and the Ag nanoparticles were dried in a vacuum oven overnight at 80 °C. The sample compositions are listed in Table 1.

### 2.4. Injection Molding

The composite-testing samples of POM, POM/Ag, and POM/Ag/MAH-*g*-POE nanocomposites were produced by using an injection molding machine (ZSJ-5-C, Shanghai, China) at a melting temperature of 220 °C, injection pressure of 0.5 MPa, and holding time of 10 s in the mold.

### 2.5. Fourier Transform Infrared Spectroscopy (FTIR)

FTIR measurements were performed on an infrared spectrometer (Spectrum 100, PerkinElmer, Shanghai, China). All spectra were recorded in the transmission mode in the 500–4000 cm^−1^ region.

### 2.6. Scanning Electron Microscopy (SEM)

Morphological and microstructural features of POM, POM/Ag, and POM/Ag/MAH-*g*-POE nanocomposites were investigated by SEM (Quanta450, FEI, Hillsboro, OR, USA). The cross-sectioned samples were prepared by fracturing the samples in liquid nitrogen. Next, the cross-sectioned samples were adequately etched in xylene for 4 h at 52 °C to remove amorphous MAH-*g*-POE and were sputter-coated with gold powder [29].

### 2.7. Thermogravimetric Analysis (TGA)

Thermal stability experiments were performed by using a TG analyzer (Pyris I, PerkinElmer, Shanghai, China) under a nitrogen environment at a heating rate of 10 °C/min from 25 to 650 °C.

### 2.8. Differential Scanning Calorimetry (DSC)

The DSC measurements were performed on a DSC 204F1(Netzsch, Selb, Germany). Samples weighing about 5–8 mg in an aluminum crucible were heated from 30 to 200 °C with a heating rate of 10 °C/min and were kept at 200 °C for 5 min to remove the thermal history, and they were then cooled from 200 °C to 30 °C with a cooling rate of 10 °C/min. After completion of the melt-crystallization process, samples were kept at 30 °C for 5 min. Then, the samples were heated again from 30 °C to 200 °C with a heating rate of 10 °C/min. In addition, the second melting and the first cooling curves were recorded.

The degree of crystallinity (*X_C_*) was calculated by Equation (1) [30]:(1)$$X_{C}(\%)=\frac{\Delta H_{m}}{(1-\alpha )\Delta H_{m}^{0}}\times 100$$
where Δ*H_m_* is the enthalpy of melting in the second heating scan of the samples (J/g), $\Delta H_{m}^{0}$ is the enthalpy value of melting of a 100% crystalline form of matrix polymer, and α is the weight fraction of Ag nanoparticles. The $\Delta H_{m}^{0}$ value of POM was 326 J/g [31].

### 2.9. Mechanical Testing

The tensile strength and elongation at break of nanocomposites were determined on a universal testing machine (Zwick Z100, Ulm, Germany) at room temperature according to the GB/T1040.1-2006 standard. The notched impact strength tests were performed according to GB1840-80 at room temperature in an impact tester (PTM7151-C, Shanghai, China). Each sample was tested 5 times and the average value was taken.

### 2.10. Dynamic Mechanical Analysis (DMA)

The dynamic mechanical thermal analysis of nanocomposites was measured on a dynamic mechanical analyzer (NETZSCH 242C, Selb, Germany) within a temperature range from −120 °C to 155 °C in nitrogen, at a heating rate of 3 °C/min, at a frequency of 1 Hz, and at a deformation of 0.1%.

## 3. Results and Discussion

### 3.1. FTIRAnalysis

The FTIR spectra of all samples were shown in Figure 2. In curve a of Figure 2, the broad and strong band at 1562 cmcm^−1^ was due to the asymmetric vibration of –COO^−^, and the 3434 cm^−1^ absorption peak was attributed to the bonding of N–H of n-propylamineon the Ag nanoparticles surface, which indicates that the complex of oleic acid and n-propylamine was successfully coated on the surface of the Ag nanoparticles. In the spectrum of POM (curve b), the characteristic peaks of POM were observed at 881, 1080, and 1230 cmcm^−1^ (stretching vibration of C–O–C groups), and 2740–3024 cm^−1^(stretching vibration of C–H group).Meanwhile, the3434cm^−1^ absorption peak (stretching vibration of N–H), the 2740–3024 cm^−1^ absorption peak (stretching vibration of C–H group), and the 881, 1080, and 1230 cm^−1^ absorption peaks (stretching vibration of C–O–C groups) were also observed on the spectra of the POM/Ag nanocomposites (curve c)and POM/Ag/MAH-*g*-POE nanocomposites (curve d), which were associated with the Ag nanoparticles (curve a) and POM (curve b). In addition, in the spectrum of POM/Ag/MAH-*g*-POE nanocomposites (curve d), the peak at 1725 cm^−1^ corresponded to the stretching vibration of C=O of MAH-*g*-POE.

### 3.2. SEManalysis

SEM images of POM/Ag/MAH-*g*-POE nanocomposites are shown in Figure 3. Meanwhile, SEM and TEM images of the POM/Ag nanocomposites can be found in the previously published papers [27,28]. Figure 3a shows the SEM image of POM, and it can be seen that there were no nanoparticles in the POM matrix, and the surface was smooth. However, an obvious multiphase structure could be observed after removing the MAH-*g*-POE-dispersed phase through the etching of xylene, where the holes represent the removed MAH-*g*-POE component, as shown in Figure 3b–f. We suggest that the MAH-*g*-POE plays a role in the bridging of the Ag nanoparticles and POM, which improved the interfacial adhesion between the Ag nanoparticles and POM matrix. In addition, with the increase in Ag/MAH-*g*-POE content from 1 to 20 wt.%, the number and size of pores in POM/Ag/MAH-*g*-POE nanocomposites increased, and the pores were well dispersed in the POM matrix, which indicates that Ag/MAH-*g*-POE had a good compatibility with POM.

### 3.3. TGAanalysis

The thermal stabilities of POM/Ag nanocomposites and POM/Ag/MAH-*g*-POE nano composites were evaluated by the TGA method. The TGA and DTG diagrams of POM/Ag nano composites are shown in Figure 4. It can be found from Figure 4a,b that POM had only one thermal decomposition interval, which mainly occurred in the temperature range of 280–350 °C. In addition, the maximum decomposition rate temperature was 326.29 °C, and POM almost completely decomposed when the temperature rose to 350 °C. In addition, with the increase in content of Ag nanoparticles, the initial decomposition temperature, maximum decomposition rate temperature, and final decomposition temperature of POM/Ag nanocomposites increased. When the content of Ag nanoparticles increased to 2 wt.%, the initial decomposition temperature of POM/Ag nanocomposites increased to 310 °C, and the maximum decomposition rate temperature increased to 367.29°C, which were 30 and 41°C higher than those of POM, respectively. It can be seen that Ag nanoparticles increased the thermal stability of POM, because there was a strong interaction between Ag nanoparticles and POM, and Ag nanoparticles inhibited the chain breaking of POM [32,33]. Additionally, the initial decomposition temperature, maximum decomposition rate temperature, and final decomposition temperature of POM/Ag/MAH-*g*-POE ternary nanocomposites also significantly increased with the Ag/MAH-*g*-POE content. It was clear that the MAH-*g*-POE also improved the thermal stability of the POM/Ag/MAH-*g*-POE ternary nanocomposites, due to the good compatibility and dispersion. Furthermore, the initial decomposition temperature and the maximum decomposition rate temperature of MAH-*g*-POE were 401 and 448°C, respectively [34]. When the Ag/MAH-*g*-POE content was 20 wt.%, the maximum decomposition temperature of POM/Ag/MAH-*g*-POE ternary nanocomposites was 398.22 °C, which was 71.39 °C higher than that of pure POM. Meanwhile, POM/Ag/MAH-*g*-POE ternary nanocomposites had a relatively small weight loss period at 475 °C, which was mainly the decomposition of MAH-*g*-POE [35]. With the increase in Ag/MAH-*g*-POE content, the decomposition stage range of MAH-*g*-POE increased. MAH-*g*-POE and Ag nanoparticles had a strong interaction with POM, which inhibited the chain breaking of POM and improved the activation energy of POM thermal degradation. Overall, both Ag nanoparticles and MAH-*g*-POE could enhance the thermal stability of the POM/Ag/MAH-*g*-POE ternary nanocomposite.

### 3.4. DSC Analysis

The melting and crystallization behavior of POM/Ag/MAH-*g*-POE ternary nanocomposites are shown in Figure 5 and the DSC parameters are given in Table 2. Meanwhile, the melting and crystallization behavior of the POM/Ag nanocomposites can be found in previously published papers [27,28]. As seen in Figure 5, there were no significant changes in the melting peak temperature (*T_m_*) and the crystallization peak temperature (*T**_c_*) values of POM/Ag/MAH-*g*-POE ternary nanocomposites. Comparing with POM, the melting peak and crystallization peak of 1P-A-M became sharp, and the degree of crystallinity (*X_C_*) value increased to 48.86%due to the heterogeneous nucleation effect of Ag nanoparticles, when the amount of Ag/MAH-*g*-POE was low (≤1 wt.%). However, with the increase in Ag/MAH-*g*-POE content, the melting and crystallization peaks of POM/Ag/MAH-*g*-POE nanocomposites became shorter and wider, and the *X_C_* of POM/Ag/MAH-*g*-POE nanocomposites decreased continuously. In addition, the *X_C_* of 5P-A-Mwas smallest, at 38.07%. When the Ag/MAH-*g*-POE content was higher than 1 wt.%, the interaction between MAH-*g*-POE macromolecular chains with POM macromolecular chains hindered the regular arrangement of POM macromolecular chains, which inhibited the POM crystallization process and reduced the degree of crystallinity of POM.

### 3.5. Mechanical Properties

The mechanical properties of POM/Ag nanocomposites are shown in Figure 6a,b. Meanwhile, the mechanical properties data of POM/Ag nanocomposites are listed in Table 3. The tensile strength, elongation at break, and Young’s modulus of POM were 50.03 MPa, 48.05%, and 1293.91MPa, respectively. However, the tensile strength and Young’s modulus of POM/Ag nanocomposites first increased and then decreased with the increase in Ag nanoparticles content. When the Ag nanoparticles content was 1 wt.%, the tensile strength and Young’s modulus reached maximum values of 56.12 and 1450.39 MPa, respectively. On the contrary, the elongation at break of POM/Ag nanocomposites first decreased and then increased with the Ag nanoparticles content. When the Ag nanoparticles content was 1 wt.%, the elongation at break reached the minimum value of 43.94%. Ag nanoparticles played the role of physical cross-linking points in POM. That is, when a molecular chain of POM is stressed, it is transferred to other molecular chains through the cross-linking point [36]. However, with the increase in Ag nanoparticles content, the excessive Ag nanoparticles were easy to agglomerate and become the defects of POM/Ag nanocomposites, resulting in the decrease in tensile strength and Young’s modulus [37,38]. In addition, Ag nanoparticles increased the crystallinity of POM, resulting in the decrease in elongation at break of POM/Ag nanocomposites. Additionally, the impact strength of POM/Ag nanocomposites decreased with the increase in Ag nanoparticles content, which was attributed to Ag nanoparticles increasing the crystallinity of POM and reducing the toughness of POM.

The mechanical properties of POM/Ag/MAH-*g*-POE ternary nanocomposites are shown in Figure 6c,d. Meanwhile, the mechanical properties of POM/Ag/MAH-*g*-POE nanocomposites are also listed in Table 3. The results showed that with the increase in Ag/MAH-*g*-POE content, the tensile strength and Young’s modulus decreased. When the Ag/MAH-*g*-POE content was 1 wt.%, the tensile strength and Young’s modulus reached maximum values of 54.78 and 1415.76 MPa, respectively. However, the elongation at break of POM/Ag/MAH-*g*-POE ternary nanocomposites increased at first and then decreased. When the Ag/MAH-*g*-POE content increased to 15wt.%, the elongation at break reached the maximum value of 64.02% due to the continuous band phase of the MAH-*g*-POE elastomer in POM, which absorbed the stress of POM. Additionally, when the Ag/MAH-*g*-POE content further increased to 20 wt.%, the elongation at break decreased again, which could be attributed to the aggregation of excessive Ag nanoparticles forming local defects in the POM/Ag/MAH-*g*-POE ternary nanocomposites. Figure 6d shows the impact strength of POM/Ag/MAH-*g*-POE ternary nanocomposites with the change in Ag/MAH-*g*-POE content. According to Table 3, the impact strength of POM was 5.57 kJ/m^2^. However, the impact strength of POM/Ag/MAH-*g*-POE ternary nanocomposites was larger than that of POM. When the Ag/MAH-*g*-POE content increased to 15 wt.%, the impact strength of POM/Ag/MAH-*g*-POE nanocomposites reached the maximum value of 8.98 kJ/m^2^, which was 1.6 times that of POM.

In order to verify the toughening effect of the MAH-*g*-POE elastomer on POM, 1P-A-M and 5P-A-M were selected and analyzed by SEM. It can be found from Figure 7a that when the Ag/MAH-*g*-POE content was 1 wt.%, the impact section of 1P-A-M was an obvious concave convex undulation, and the undulation boundaries and the boundary segmentation lines were obvious, which was mainly brittle fracture. However, when the Ag/MAH-*g*-POE content increased to 20 wt.%, the crack on the impact section of 5P-A-M was obvious, indicating that plastic deformation and shear yield of the POM/Ag/MAH-*g*-POE nanocomposites occurred. At the same time, the shear band could control the crack developing into a destructive crack, and the stress field at the crack tip could induce a shear band, which consumed a lot of energy. It was proved that the MAH-*g*-POE elastomer could significantly enhance the toughness of POM.

### 3.6. DMA Analysis

Figure 8a,b show the dynamic mechanical properties of POM/Ag nanocomposites versus the content of Ag nanoparticles. It was found that, with the increase in the content of Ag nanoparticles, the storage modulus of POM/Ag nanocomposites showed an increasing trend due to the stiffness of Ag nanoparticles, which showed that the rigidity of POM/Ag nanocomposites increased [39]. The loss modulus of POM/Ag nanocomposites corresponded to the energy loss caused by the interaction between Ag nanoparticles and POM macromolecular chains and the friction between Ag nanoparticles and POM macromolecular chains. It can be seen from Figure 8b that the loss modulus of POM/Ag nanocomposites increased significantly with increased Ag nanoparticles. At –100 °C, when the content of Ag nanoparticles increased at 2 wt.%, the storage modulus of POM/Ag nanocomposites increased to 5956.44 MPa, which increased by 35.69%. In addition, Figure 8c,d show the curve of dynamic mechanical properties of POM/Ag/MAH-*g*-POE ternary nanocomposites with different Ag/MAH-*g*-POE contents. The results showed that, compared with POM, the storage modulus of POM/Ag/MAH-*g*-POE ternary nanocomposites decreased with the Ag/MAH-*g*-POE content, which could be due to the MAH-*g*-POE elastomer reducing the rigidity of POM. It can be found from Figure 8d that the peak value of the loss modulus of POM/Ag/MAH-*g*-POE ternary nanocomposites decreased significantly near the glass transition temperature (−53 °C) of POM. The MAH-*g*-POE could increase the elasticity of POM by transferring a uniform external force. In addition, when the Ag/MAH-*g*-POE content was lower than 20 wt.%, there was one peak at −53 °C, which proved that Ag/MAH-*g*-POE had good compatibility with POM. However, when the Ag/MAH-*g*-POE content increased to 20 wt.%, there were two peaks near the glass transition temperature, which indicates that microscopic phase separation formed in the POM/Ag/MAH-*g*-POE ternary nanocomposites.

## 4. Conclusions

POM/Ag/MAH-*g*-POE ternary nanocomposites with varying contents of Ag nanoparticles and MAH-*g*-POE were prepared by a melt mixing method, and the effects of the Ag nanoparticles and MAH-*g*-POE contents on the microstructure, thermal stability, crystallization behavior, and mechanical properties of the nanocomposites were studied. When the content of Ag/MAH-*g*-POE was less than 20 wt.%, the dispersion effect of Ag/MAH-*g*-POE in POM was very good, indicating that the compatibility between Ag/MAH-*g*-POE and the POM matrix was good. In addition, Ag nanoparticles and Ag/MAH-*g*-POE could significantly enhance the thermal stability of POM/Ag/MAH-*g*-POE ternary nanocomposites. The maximum decomposition rate temperature of POM/Ag nanocomposites increased to 367.29°C, which was 41°C higher than that of POM, and the maximum decomposition temperature of POM/Ag/MAH-*g*-POE nanocomposites was 398.22 °C, which was 71.39 °C higher than that of POM. In addition, the *X_C_* value of POM/Ag/MAH-*g*-POE nanocomposites increased to 48.86%due to the heterogeneous nucleation effect of Ag nanoparticles, when the amount of Ag/MAH-*g*-POE was low (≤1 wt.%). Meanwhile, when the Ag nanoparticles content was 1 wt.%, the tensile strength of POM/Ag nanocomposites reached the maximum value of 56.12 MPa and the elongation at break reached the minimum value of 43.94%. At the same time, when the Ag/MAH-*g*-POE content was 1 wt.%, the tensile strength of POM/Ag/MAH-*g*-POE nanocomposites reached the maximum value of 54.78 MPa. However, when the Ag/MAH-*g*-POE content increased to 15wt.%, the elongation at break of POM/Ag/MAH-*g*-POE nanocomposites reached the maximum value of 64.02% and the impact strength reached the maximum value of 8.98 kJ/m^2^ due to the continuous band phase of the MAH-*g*-POE elastomer in POM, which absorbed the stress of POM. Furthermore, with the increase in the content of Ag nanoparticles, the storage modulus of POM/Ag nanocomposites showed an increasing trend due to the stiffness of Ag nanoparticles. However, the loss modulus of POM/Ag/MAH-*g*-POE nanocomposites decreased with the Ag/MAH-*g*-POE content, because the MAH-*g*-POE elastomer could reduce the rigidity of the POM matrix.

## Figures and Tables

**Figure 1 polymers-13-01954-f001:**
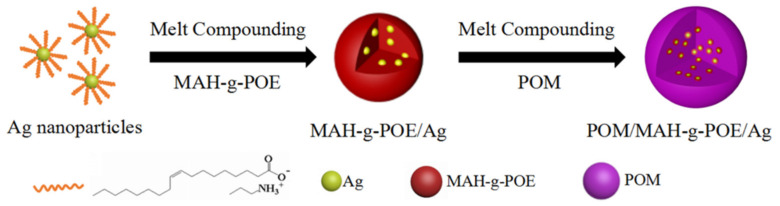
Synthesis procedure of POM/Ag/MAH-*g*-POE nanocomposites.

**Figure 2 polymers-13-01954-f002:**
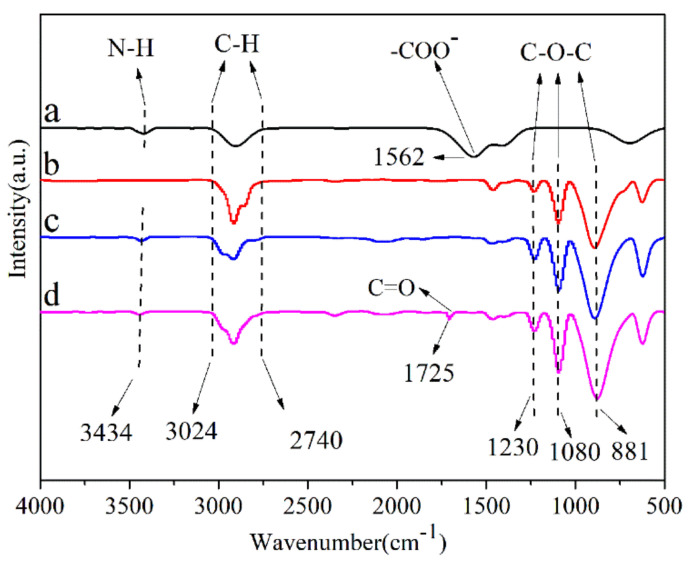
FTIR spectra of the (**a**) Ag nanoparticles, (**b**) POM, (**c**) POM/Ag nanocomposites, and (**d**) POM/Ag/MAH-*g*-POE nanocomposites.

**Figure 3 polymers-13-01954-f003:**
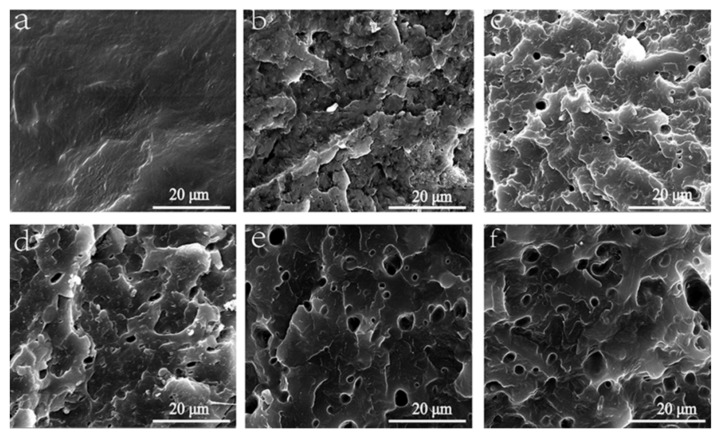
SEM images of the (**a**) POM, (**b**) 1P-A-M, (**c**) 2P-A-M, (**d**) 3P-A-M, (**e**) 4P-A-M, and (**f**) 5P-A-M.

**Figure 4 polymers-13-01954-f004:**
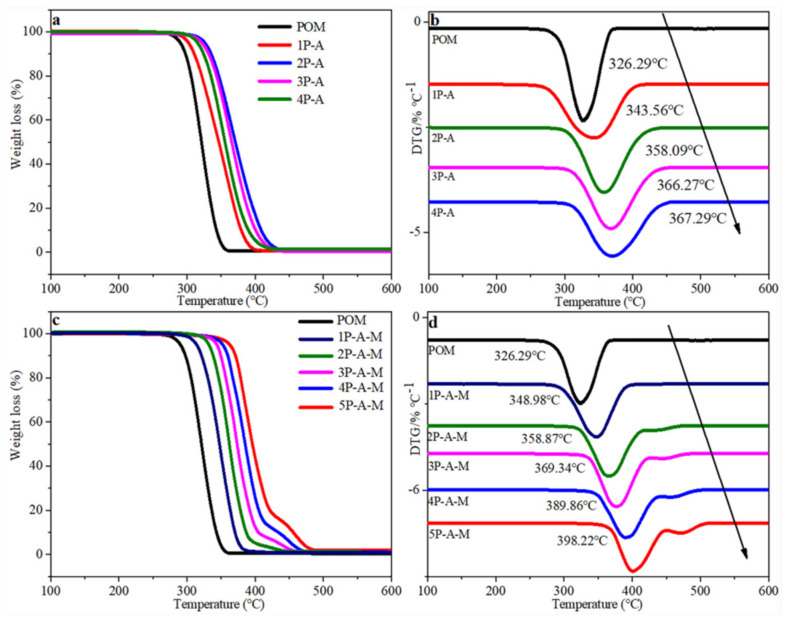
(**a**) TGA and (**b**) DTG diagrams of POM/Ag nanocomposites; (**c**) TGA and (**d**) DTG diagrams of POM/Ag/MAH-*g*-POE nanocomposites.

**Figure 5 polymers-13-01954-f005:**
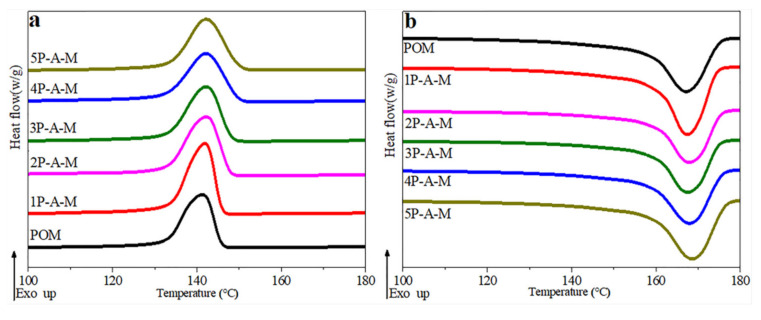
(**a**) Crystallization exotherms and (**b**) second melting end other ms of POM/Ag/MAH-*g*-POE ternary nanocomposites at a heating/cooling rate of 10 °C/min.

**Figure 6 polymers-13-01954-f006:**
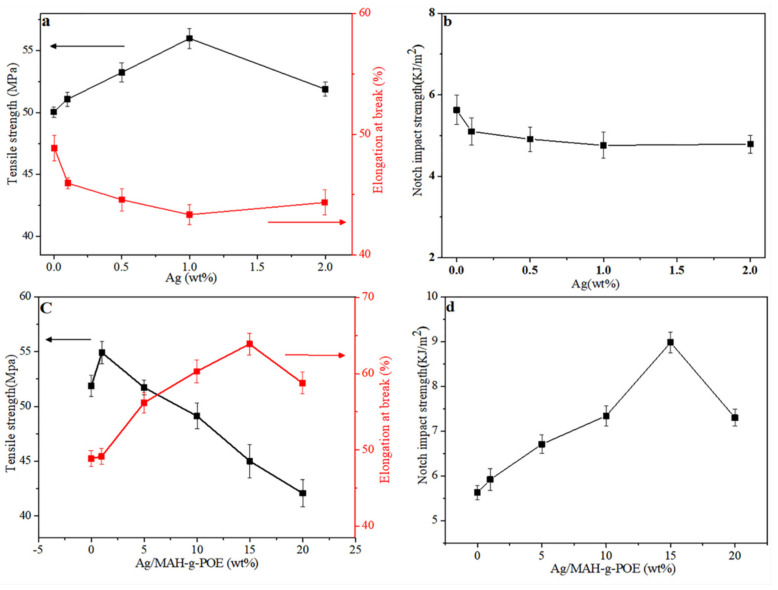
(**a**) Tensile strength and elongation at break and (**b**) impact strength of POM/Ag nanocomposites; (**c**) tensile strength and elongation at break and (**d**) impact strength of POM/Ag/MAH-*g*-POE ternary nanocomposites.

**Figure 7 polymers-13-01954-f007:**
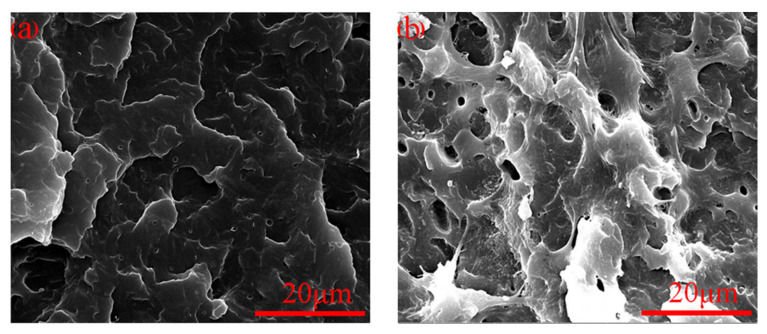
SEM of impact section of (**a**) 1P-A-M and (**b**) 5P-A-M.

**Figure 8 polymers-13-01954-f008:**
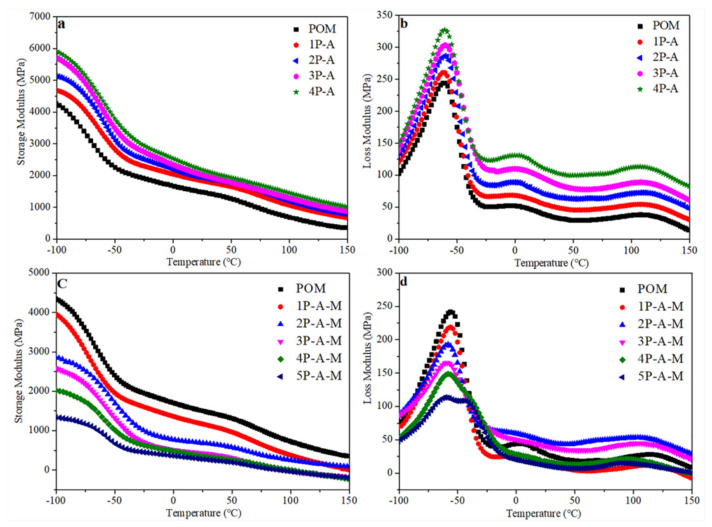
(**a**)Storage modulus diagrams and (**b**) loss modulus diagrams of POM/Ag nanocomposites; (**c**) storage modulus diagrams and (**d**) loss modulus diagrams of POM/Ag/MAH-*g*-POE ternary nanocomposites.

**Table 1 polymers-13-01954-t001:** Formulations of the nanocomposites.

Sample Code (Abbreviation)	POM (wt.%)	Ag (wt.%)	MAH-*g*-POE (wt.%)
POM	100	0	0
1P-A	99.9	0.1	0
2P-A	99.5	0.5	0
3P-A	99	1	0
4P-A	98	2	0
1P-A-M	99	0.1	0.9
2P-A-M	95	0.5	4.5
3P-A-M	90	1	9
4P-A-M	85	1.5	13.5
5P-A-M	80	2	18

**Table 2 polymers-13-01954-t002:** Characteristic crystallization peak temperatures. Enthalpy and degree of crystallinity values of samples crystallized with a cooling rate of 10 °C/min.

Samples	*T_m_* (°C)	*T_c_* (°C)	Δ*H_m_* (J/g) ^a^	*X_C_* (%) ^b^
POM	168.24	140.64	152.94	46.91
1P-A-M	168.83	140.83	159.29	48.86
2P-A-M	167.91	141.81	143.89	44.13
3P-A-M	167.23	142.13	138.64	42.52
4P-A-M	168.81	140.71	130.71	40.01
5P-A-M	169.31	140.15	124.11	38.07

^a^ Enthalpy of second melting endotherm recorded at the heating rate of 10 °C/min. ^b^ Degree of crystallinity calculated by Equation (1) by using the enthalpy values of second melting.

**Table 3 polymers-13-01954-t003:** Mechanical proper ties of POM/Ag nanocomposites and POM/Ag/MAH-*g*-POE ternary nanocomposites.

Samples	Tensile Strength (MPa)	Elongation at Break (%)	Young Modulus (MPa)	Notched Impact Strength (kJ/m^2^)
POM	50.03	48.05	1293.91	5.57
1P-A	51.79	46.31	1338.43	5.15
2P-A	53.18	44.88	1374.11	4.95
3P-A	56.12	43.94	1450.39	4.87
4P-A	52.27	44.92	1350.89	4.91
1P-A-M	54.78	49.78	1415.76	5.92
2P-A-M	48.69	58.17	1255.36	6.71
3P-A-M	46.62	60.66	1204.87	7.34
4P-A-M	44.92	64.02	1163.93	8.98
5P-A-M	43.27	58.72	1119.29	7.31

## Data Availability

Not applicable.

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
