# Peer review of "Structure and Properties of Polyoxymethylene/Silver/Maleic Anhydride-Grafted Polyolefin Elastomer Ternary Nanocomposites"

_polymers, 2021, doi:10.3390/polym13121954_

Round 1

Reviewer 1 Report

The article entitled "Structure and Properties of Polyoxymethylene/Silver/Maleic Anhydride Grafted Polyolefin Elastomer Ternary Nanocomposites" by Yang Liu, Xun Zhang, Quanxin Gao, Hongliang Huang, Yongli Liu, Minghua Min, Lumin Wang describes the influence of the additives on the microstructure, dynamic thermomechanical, crystallization behaviour and tensile properties of the POM/Ag/MAH-g-POE ternary nanocomposites with varying contents of Ag nanoparticles and maleic anhydride grafted polyolefin elastomer (MAH-g-POE). Also mixing storage modulus, shear stress, loss modulus, tensile strength, elongation at break, degree of crystallinity and thermal degradation of these nanocomposites were evaluated. It was shown that the addition of Ag nanoparticles and the addition of maleic anhydride grafted polyolefin elastomer have a good influence on thermal stability and mechanical properties of POM. The paper is interesting. It is well written and can be published as is.

Author Response

Response to Reviewer 1#

Comment: The article entitled "Structure and Properties of Polyoxymethylene/Silver/Maleic Anhydride Grafted Polyolefin Elastomer Ternary Nanocomposites" by Yang Liu, Xun Zhang, Quanxin Gao, Hongliang Huang, Yongli Liu, Minghua Min, Lumin Wang describes the influence of the additives on the microstructure, dynamic thermomechanical, crystallization behaviour and tensile properties of the POM/Ag/MAH-g-POE ternary nanocomposites with varying contents of Ag nanoparticles and maleic anhydride grafted polyolefin elastomer (MAH-g-POE). Also mixing storage modulus, shear stress, loss modulus, tensile strength, elongation at break, degree of crystallinity and thermal degradation of these nanocomposites were evaluated. It was shown that the addition of Ag nanoparticles and the addition of maleic anhydride grafted polyolefin elastomer have a good influence on thermal stability and mechanical properties of POM. The paper is interesting. It is well written and can be published as is.

Response: Thank you very much for your kind comments on our manuscript.

Reviewer 2 Report

The paper reports the preparation and characterization of POM/Ag/MAH-g-POE nanocomposites. The authors analyzed the influence of the components used on the microstructure, thermal and mechanical features of the resulting samples. However, many issues are presented during the presentation and discussion of results on the present manuscript:

  • First of all, the manuscript is difficult to read and a major language correction should be performed.
  • Why did the authors use MAH-g-POE in the preparation of ternary composites? What is the advantage of this polymer over other structures that can be used as additives?
  • If the cross-sectioned samples for the SEM analyses were etched in xylene, how you can know if Ag NPs did not migrate as well from the matrix, especially if their average size was about 10 nm (you may also mention the dimension of Ag NPs in the manuscript, I searched for the size of NPs in your previous article). What do you mean by “adequately etched”? Please detail the procedure and eventually include a reference paper.
  • The sentence “It could be seen that Ag nanoparticles increased the thermal stability of POM, because there was strong interaction between Ag nanoparticles and POM, and Ag nanoparticles inhibited the chain breaking of POM.” should be sustained by a literature reference.
  • The TGA and DTG diagrams for MAH-g-POE should be included in the manuscript, to confirm the enhanced thermal stability of this component that is further translated to the ternary composites.
  • Since you included the tensile strength and elongation data in Table 3, it would be easier to observe the differences between the samples if in Figure 6 (a and c) you represent the stress-strain curves for the proposed samples. Also, an estimation of Young modulus could be of interest.
  • In Figure 6 (b) I think you want to represent (according to your comment) the impact strength of POM/Ag nanocomposites, while you put a figure illustrating tensile strength, which does not appear to be related to the data presented.
  • In Figure 7, the analyzed samples were also etched in xylene, since in the image corresponding to sample 5P-A-M some voids are visible. Is it not possible for these voids to occur as a result of the mixing process? Perhaps some annealing of the samples would help the compatibility?

Author Response

Response to Reviewer 2#

Dear reviewer,

We thank the reviewer for your careful reading of our manuscript and helpful comments.

The paper reports the preparation and characterization of POM/Ag/MAH-g-POE nanocomposites. The authors analyzed the influence of the components used on the microstructure, thermal and mechanical features of the resulting samples. However, many issues are presented during the presentation and discussion of results on the present manuscript:

  • Comment 1: First of all, the manuscript is difficult to read and a major language correction should be performed.

Response 1: Thank the reviewer. We revised the manuscript and hope it can meet the high level of the journal of Polymers.

  • Comment 2: Why did the authors use MAH-g-POE in the preparation of ternary composites? What is the advantage of this polymer over other structures that can be used as additives?

Response 2: Polyolefin elastomer (POE) has excellent toughness and good processability and the toughness of POM can be improved significantly by means of melt blending with POE. In addition, MAH can improve the dispersion of POE in a POM matrix. The MAH-g-POE, with a lone pair of electrons, is a common compatibilizer for POE based nanocomposites because of the high reactivity oxygen that exists in the MAH molecule.

  • Comment 3: If the cross-sectioned samples for the SEM analyses were etched in xylene, how you can know if Ag NPs did not migrate as well from the matrix, especially if their average size was about 10 nm (you may also mention the dimension of Ag NPs in the manuscript, I searched for the size of NPs in your previous article). What do you mean by “adequately etched”? Please detail the procedure and eventually include a reference paper.

Response 3: The xylene is a good solvent for POE-g-MAH but not for POM and Ag NPs. The operational details can be found in the following literature:

  1. Wang, B.B.; Yang, Y.; Guo, W.N. Effect of EVOH on the morphology, mechanical and barrier properties of PA6/POE-g-MAH/EVOH ternary blends. Mater. Des.2012, 40, 185-189.

So we revised the manuscript as follow:

“Next the cross-sectioned sampleswere adequately etched in xylene for 4 h at 52 °C to remove amorphous MAH-g-POE and sputter-coated with gold powder[29].”

  • Comment 4: The sentence “It could be seen that Ag nanoparticles increased the thermal stability of POM, because there was strong interaction between Ag nanoparticles and POM, and Ag nanoparticles inhibited the chain breaking of POM.” should be sustained by a literaturereference.

Response 4: We add the following literature references in the manuscript:

  1. Chae, D.W.; Shim, K.B.; Kim, B.C. Effects of silver nanoparticles on the dynamic crystallization and physical properties of syndiotactic polypropylene. J. Appl. Polym. Sci. 2008, 109, 2942-2947.
  2. Mbhele,Z.H.; Salemane, M.G.; VanSittert, C.G.C.E.; Nedeljković,J.M.; Djoković, V.; Luyt, A.S. Fabrication and characterization of silver-polyvinyl alcoholnanocomposites. Chem. Mater.2003, 15, 5019-5024.

And we revised the manuscript as follow:

“It could be seen that Ag nanoparticles increased the thermal stability of POM, because there was strong interaction between Ag nanoparticles and POM, and Ag nanoparticles inhibited the chain breaking of POM [32, 33].”

  • Comment 5: The TGA and DTG diagrams for MAH-g-POE should be included in the manuscript, to confirm the enhanced thermal stability of this component that is further translated to the ternary composites.

Response 5: The thermal behavior of MAH-g-POE was given in the reference [34]:

  1. Zhang, X.; Zhang, L.P.; Wu, Q.; Mao, Z.P. The influence of synergistic effects of hexakis(4-nitrophenoxy) cyclotriphosphazene and POE-g-MA on anti-dripping and flame retardancy of PET. J. Ind. Eng. Chem. 2013, 3, 993-999.

It was found that the initial decomposition temperature and the maximum decomposition rate temperatureof MAH-g-POE were 401°C and 448°C, respectively. In addition, the improved compatibility and dispersion could providethe composites the idealthermal stability. So we revised the manuscript as follow:

“It was clearly that the MAH-g-POE also improved the thermal stability of the POM/Ag/MAH-g-POE ternary nanocomposites, due to the good compatibility and dispersion. Furthermore, the initial decomposition temperature and the maximum decomposition rate temperatureof MAH-g-POE were 401°C and 448°C, respectively [34]. ”

  • Comment 6: Since you included the tensile strength and elongation data in Table 3, it would be easier to observe the differences between the samples if in Figure 6 (a and c) you represent the stress-strain curves for the proposed samples. Also, an estimation of Young modulus could be of interest.

Response 6: The tensile strength and elongation atbreak of nanocomposites were determined on a universal testing machine (Zwick Z100, Germany) at room temperature according tothe GB/T1040.1-2006 standard. Each nanocomposite was tested 5 times and the average value was shown in Table 3. So it was not suitable to represent the stress-strain curves for all the samples. In addition, we added the Young modulus of POM/Ag nanocomposites and POM/Ag/MAH-g-POE nanocomposites in the Table 3. And the Table 3 was revised as follow:

Table 3. Mechanical properties of POM/Ag nanocomposites and POM/Ag/MAH-g-POE ternary nanocomposites.

Samples

Tensile strength (MPa)

Elongation at break (%)

Young modulus(MPa)

Notched impact strength(kJ/m2)

POM

50.03

48.05

1293.91

5.57

1P-A

51.79

46.31

1338.43

5.15

2P-A

53.18

44.88

1374.11

4.95

3P-A

56.12

43.94

1450.39

4.87

4P-A

52.27

44.92

1350.89

4.91

1P-A-M

54.78

49.78

1415.76

5.92

2P-A-M

48.69

58.17

1255.36

6.71

3P-A-M

46.62

60.66

1204.87

7.34

4P-A-M

44.92

64.02

1163.93

8.98

5P-A-M

43.27

58.72

1119.29

7.31

  • Comment 7: In Figure 6 (b) I think you want to represent (according to your comment) the impact strength of POM/Ag nanocomposites, while you put a figure illustrating tensile strength, which does not appear to be related to the data presented.

Response 7: We replaced tensile strength with impact strength in Figure 6(b)and the Figure 6 was revised in the manuscript as follow:

Figure 6. (a) Tensile strength and elongation at break and (b) impact strength of POM/Ag nanocomposites; (c)tensile strength and elongation at break and(d) impact strength of POM/Ag/MAH-g-POE ternary nanocomposites.

  • Comment 8: In Figure 7, the analyzed samples were also etched in xylene, since in the image corresponding to sample 5P-A-M some voids are visible. Is it not possible for these voids to occur as a result of the mixing process? Perhaps some annealing of the samples would help the compatibility?

Response 8: SEM of impact section of 5P-A-M before etched by xylene was shown as follow:

Figure 1: SEM of impact section of 5P-A-Mbefore etched by xylene.

It was found that the sample 5P-A-M before etched by xylene has no discernible voids. The ‘some annealing of the samples would help the compatibility’ might be good advice, and we’ll try this method in the near future.

Round 2

Reviewer 2 Report

The problems identified were mainly solved by the authors, although some explanations could have been introduced in the manuscript (for example in comment 2). However, the manuscript does not appear to have been corrected by a person with better English language expertise. At least, please replace "silver nitric" in the manuscript with "silver nitrate", which is the IUPAC name for AgNO3. Also, the sentence "The influence of the additives on the microstructure, dynamic thermo-mechanical, crystallization behavior and tensile properties of the nanocomposites was evaluated, mixing storage modulus, shear stress, loss modulus, tensile strength, elongation at break,degree of crystallinity and thermal degradation of the nanocomposites." which appears both in the abstract and in the introduction, needs to be clarified. Also, on line 82, the word rapture has to be replaced by rupture (I guess).

Author Response

Response to Reviewer2#

Dear Reviewer,

We would like to express our great appreciation to the reviewer 2# for the comments on our manuscript, and we have made corresponding revisionin the manuscript.

Comments: The problems identified were mainly solved by the authors, although some explanations could have been introduced in the manuscript (for example in comment 2). However, the manuscript does not appear to have been corrected by a person with better English language expertise. At least, please replace "silver nitric" in the manuscript with "silver nitrate", which is the IUPAC name for AgNO3. Also, the sentence "The influence of the additives on the microstructure, dynamic thermo-mechanical, crystallization behavior and tensile properties of the nanocomposites was evaluated, mixing storage modulus, shear stress, loss modulus, tensile strength, elongation at break,degree of crystallinity and thermal degradation of the nanocomposites." which appears both in the abstract and in the introduction, needs to be clarified. Also, on line 82, the word rapture has to be replaced by rupture (I guess).

Response: We carefully revised our manuscript, and some detailed changes were listed below:

  1. In the Abstract, the sentence “Polyoxymethylene (POM) is a semicrystalline thermoplastic and its widespread application is limited by its low elongation at break and thermal durability” was deleted.
  2. In the Abstract, the sentence “The influence of the additives on the microstructure, dynamic thermo-mechanical, crystallization behavior and tensile properties of the nanocomposites was evaluated, mixing storage modulus, shear stress, loss modulus, tensile strength, elongation at break, degree of crystallinity and thermal degradation of the nanocomposites.” was simplified as “The effects of the additives on the microstructure, thermal stability, crystallization behavior, mechanical properties and dynamic mechanical thermal properties of the ternary nanocomposites were studied.
  3. In the Abstract, the sentence“Moreover, it was found that the combined addition of Ag nanoparticles and MAH-g-POE enhanced the thermal stability, tensile strength, elongation at break and chemical stability of the POM/Ag/MAH-g-POE nanocomposites.” was revised as “Moreover, it was found that the combined addition of Ag nanoparticles and MAH-g-POE significantly enhanced the thermal stability, crystallization properties and mechanical properties of the POM/Ag/MAH-g-POE ternary nanocomposites.
  4. The word “rapture”has to be replaced by “rupture”.
  5. In the end of Introduction, the sentence “The influence of the additives on the microstructure, dynamic thermo-mechanical, crystallization behavior and tensile properties of the nanocomposites was evaluated, mixing storage modulus, shear stress, loss modulus, tensile strength, elongation at break, degree of crystallinity and thermal degradation of the nanocomposites.” was also simplified as “The effects of the additives on the microstructure, thermal stability, crystallization behavior, mechanical properties and dynamic mechanical thermal properties of the ternary nanocomposites were studied.
  6. In the 2.1. materials, “silver nitric” was substituted with “silver nitrate”.
  7. In the 3.3. TGA analysis, the sentence “However, the initial decomposition temperature, maximum decomposition rate temper-ature and final decomposition temperature of POM/Ag nanocomposites were higher than those of POM, increasing with the content of Ag nanoparticles increased.” was revised as “In addition, with the content of Ag nanoparticles increased, the initial decomposition temperature, maximum decomposition rate temperature and final decomposition temperature of POM/Ag nanocomposites increased.”
  8. In the end of 3.3. TGA analysis,the sentence “Furthermore, MAH-g-POE and Ag nanoparticles of POM/Ag/MAH-g-POE ternary nanocomposites prevented the volatiles from the matrix.” was revised as “All in all, both Ag nanoparticles and MAH-g-POE could enhance the thermal stability of POM/Ag/MAH-g-POE ternary nanocomposite.
  9. In the 3.4. DSC analysis, the sentence “The melting and crystallization behavior of POM/Ag/MAH-g-POE ternary nano-composites at the heating/cooling rate of 10°C/min were shown in Figure 5 and the DSC parameters were given in Table 2.” was simplified as “The melting and crystallization behavior of POM/Ag/MAH-g-POE ternary nano-composites were shown in Figure 5 and the DSC parameters were given in Table 2.”
  10. In the 3.4. DSC analysis, the sentence “When the Ag/MAH-g-POE content was high, the interaction between MAH-g-POE macromolecular chains with POM macromolecular chains hindered the regular arrangement of POM macromolecular chains, which inhibited the POM crystallization process and reduced the degree of crystallinity of POM.” was revised as “When the Ag/MAH-g-POE content was higher than 1 wt%, the interaction between MAH-g-POE macromolecular chains with POM macromolecular chains hindered the regular arrangement of POM macromolecular chains, which inhibited the POM crystallization process and reduced the degree of crystallinity of POM.
  11. In the 3.5. Mechanical properties, the sentence “Additionally, the impact strength of POM was 5.57 kJ/m2. The impact strength of POM/Ag nanocomposites first decreased and then increased with the increase of Ag nanoparticles content, and the minimum value is 4.87 kJ/m2, when the content of Ag nanoparticles was 1 wt%. It indicated that Ag nanoparticles could not improve the toughness of POM,because Ag nanoparticles increased the crystallinity of POM and reduced the toughness of POM.” was simplified as “Additionally, the impact strength of POM/Ag nanocomposites decreased with the increase of Ag nanoparticles content, which was attributed to that Ag nanoparticles increased the crystallinity of POM and reduced the toughness of POM.
  12. In the 3.5. Mechanical properties, the sentence “Additionally, when the Ag/MAH-g-POE content increased to 20 wt%, the elongation at break decreased again, because excessive MAH-g-POE was prone to molecular chain entanglement and aggregation of excessive Ag nanoparticles, resulting in local defects of POM/Ag/MAH-g-POE ternary nanocomposites.” was revised as “Additionally, when the Ag/MAH-g-POE content further increased to 20 wt%, the elongation at break decreased again, which could be attributed to that the aggregation of excessive Ag nanoparticles formed local defects in the POM/Ag/MAH-g-POE ternary nanocomposites.
  13. In the end of 3.5. Mechanical properties, the sentence “It proved that MAH-g-POE elastomer could increase the toughness of POM.” was revised as “It was proved that MAH-g-POE elastomer could significantly enhance the toughness of POM.
  14. In the 3.6. DMA analysis, the sentence “The results indicated that the storage modulus of POM/Ag nanocomposites increased, that was, the stiffness of POM/Ag nanocomposites increased. In addition, with the increased of the content of Ag nanoparticles, the storage modulus of POM/Ag nano-composites showed an increasing trend due to the stiffness of Ag nanoparticles, which showed that the rigidity of POM/Ag nanocomposites increased [39].” was revised as “It was found that, with the increased of the content of Ag nanoparticles, the storage modulus of POM/Ag nanocomposites showed an increasing trend due to the stiffness of Ag nanoparticles, which showed that the rigidity of POM/Ag nanocomposites increased [39].
  15. In the end of 3.6. DMA analysis, the sentence “Moreover, it was found from Figure 8d that when the Ag/MAH-g-POE content increased to 20 wt%, there were two peaks, which indicated that the compatibility between Ag/MAH-g-POE and POM was better when the Ag/MAH-g-POE was lower than 20 wt%.” was rewritten as “In addition, when the Ag/MAH-g-POE content lower than 20 wt%, there was one peak at -53 °C, which proved that Ag/MAH-g-POE had good compatibility with POM. However, when the Ag/MAH-g-POE content increased to 20 wt%, there were two peaks near the glass transition temperature, which indicated microscopic phase separation formed in the POM/Ag/MAH-g-POE ternary nanocomposites.
  16. The “4. Conclusions” was also revised.